# UniMatch: Universal Matching from Atom to Task for Few-Shot Drug Discovery

**Ruifeng Li**[1,2]**, Mingqian Li**[2]**, Wei Liu**[3]**, Yuhua Zhou**[1,2]**, Xiangxin Zhou**[4]**,**
**Yuan Yao**[1,5]**, Qiang Zhang**[1,5†]**, Hongyang Chen**[2†]

[1] Zhejiang University [2] Zhejiang Lab [3] Shanghai Jiao Tong University
[4] University of Chinese Academy of Sciences
[5] ZJU-Hangzhou Global Scientific and Technological Innovation Center
{lirf, qiang.zhang.cs}@zju.edu.cn, dr.h.chen@ieee.org

## ABSTRACT

Drug discovery is crucial for identifying candidate drugs for various diseases. However, its low success rate often results in a scarcity of annotations, posing a few-shot learning problem. Existing methods primarily focus on single-scale features, overlooking the hierarchical molecular structures that determine different molecular properties. To address these issues, we introduce **Uni**versal **Match**ing Networks (**UniMatch**), a dual matching framework that integrates explicit hierarchical molecular matching with implicit task-level matching via meta-learning, bridging multi-level molecular representations and task-level generalization. Specifically, our approach explicitly captures structural features across multiple levels—atoms, substructures, and molecules—via hierarchical pooling and matching, facilitating precise molecular representation and comparison. Additionally, we employ a meta-learning strategy for implicit task-level matching, allowing the model to capture shared patterns across tasks and quickly adapt to new ones. This unified matching framework ensures effective molecular alignment while leveraging shared meta-knowledge for fast adaptation. Our experimental results demonstrate that UniMatch outperforms state-of-the-art methods on the MoleculeNet and FS-Mol benchmarks, achieving improvements of 2.87% in AUROC and 6.52% in $\Delta$AUPRC. UniMatch also shows excellent generalization ability on the Meta-MolNet benchmark. The code is available at https://github.com/Lirain21/UniMatch.git

## 1 INTRODUCTION

Drug discovery is pivotal for human health, involving the screening and optimization of numerous compounds to identify potential drug candidates that satisfy both pharmacological efficacy and toxicological safety criteria (Drews, 2000; Renaud et al., 2016; Atanasov et al., 2021). The traditional drug development cycle typically spans over a decade, incurs costs exceeding $1 billion, yet achieves a success rate of less than 10% (Sliwoski et al., 2014; Adelusi et al., 2022). Artificial Intelligence-Driven Drug Discovery (AIDD) has emerged as a promising solution to address this challenge (Mak et al., 2023; Macalino et al., 2015; Gawehn et al., 2016). Within AIDD, Quantitative Structure-Activity/Property Relationships (QSAR/QSPR) (Cherkasov et al., 2014; Liu & Long, 2009) models are crucial for predicting the relationships between molecular structures and their activities. These methods rely heavily on extensive datasets due to the complexity of understanding and modeling molecular geometries (Zhang et al., 2021a; Fabian et al., 2020; Wang et al., 2021; Chen et al., 2023). However, the lengthy durations, high costs, and low success rates of chemical wet experiments limit the availability of labeled experimental data.

Few-shot learning (Li et al., 2023; Song et al., 2023) has demonstrated substantial potential in addressing data scarcity by enabling models to generalize rapidly from minimal data to new tasks. Recently, several approaches have been proposed to address this challenge in few-shot scenarios (Wang et al., 2020). Most approaches are based on molecular graphs with atoms as nodes and chemical bonds as edges, leveraging Graph Neural Networks (GNNs) (Zhou et al., 2020) to capture molecular topologies. In particular, such models as IterRefLSTM (Altae-Tran et al., 2017),

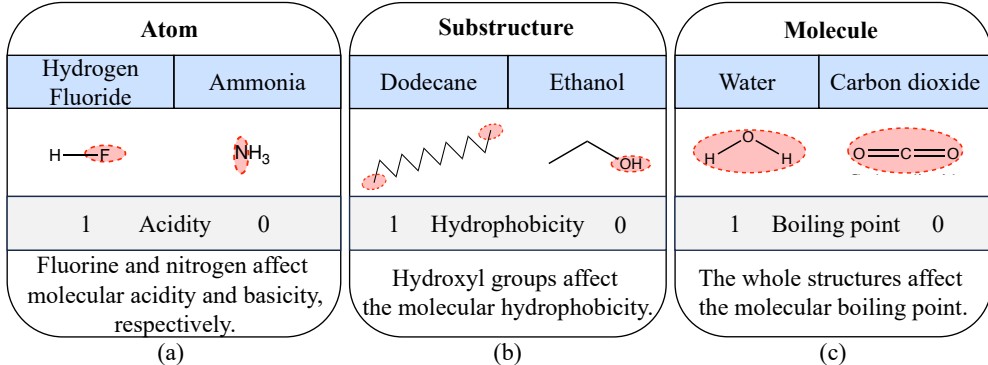

Figure 1: Different levels of molecular structures affect different properties: (a) at the atomic level, fluorine and nitrogen affect molecular acidity and basicity, respectively; (b) at the substructural level, hydroxyl groups affect the hydrophobicity of ethanol and dodecane; and (c) at the molecular level, the overall structures influence boiling points. Key molecular structures are highlighted in red.

Meta-MGNN (Guo et al., 2021), PAR (Wang et al., 2021), ADKF-IFT (Chen et al., 2023), and Meta-GAT (Lv et al., 2024) employ GNNs as encoders to learn molecular representations for label prediction. Additionally, several sequence-based methods, such as CHEF (Adler et al., 2020), MH-Nfs (Schimunek et al., 2023) and CRA (Li et al., 2024), utilize Multilayer Perceptrons (MLPs) as encoders to compress molecular fingerprints or descriptors for predictive modeling.

However, existing approaches often overlook a crucial aspect: **different levels of structural information—ranging from atoms to substructures to the entire molecule—contribute to distinct molecular properties**. Some properties are influenced by atomic composition, while others depend on substructures or the overall molecular configuration. Figure 1 illustrates this with examples: (a) fluorine and nitrogen affect molecular acidity and basicity, respectively; (b) hydroxyl groups influence the hydrophobic properties of ethanol and dodecane; and (c) the overall molecular structure affects boiling points. In graph-based methods, the use of multiple GNN layers may cause over-smoothing, where the receptive fields of nodes expand excessively, thus obscuring substructural details (Chen et al., 2020). As a result, GNNs are more suitable for predicting properties related to the overall structure of molecules. In contrast, fingerprint-based methods rely on only fragmented local features, potentially overlooking critical information about the overall molecular structure. Although CHEF (Adler et al., 2020) introduces a representation fusion strategy, its reliance on ECFP6 (Rogers & Hahn, 2010)—which is based on fixed local features—limits its ability to capture hierarchical molecular structures. Therefore, effectively capturing different levels of molecular structures is crucial for accurately predicting a wide range of molecular properties.

To address this challenge, we propose **Universal Matching Networks** (**UniMatch**), a framework that facilitates universal matching across multiple levels—from atoms to tasks—enhancing the few-shot molecular property prediction task. Our main contributions are summarized as follows:

- To the best of our knowledge, we are pioneers to introduce a universal matching approach that spans from the atomic level to the task level. This framework employs explicit hierarchical molecular matching and implicit task-level matching at distinct levels to align molecular structures with tasks. The dual matching mechanism complements itself, forming a synergistic framework that enhances the model's adaptability and generalization across various tasks (Section 3).

- We propose an explicit hierarchical molecular matching mechanism that integrates information from atoms to higher-level structures, capturing complex molecular features. By utilizing an attention-based matching module, the model aligns representations across multiple levels, selecting the most relevant features for improved prediction (Sections 3.1).

- We incorporate a meta-learning strategy to achieve implicit task-level matching, learning shared parameters that generalize across tasks. This matching occurs at an abstract level, capturing task similarities through optimization and enabling rapid adaptation and improved generalization (Section 3.2).

- Our UniMatch outperforms state-of-the-art methods on both the MoleculeNet (Section 4.1) and FS-Mol (Section 4.2) benchmarks, achieving improvements of 2.87% in AUROC and 6.52% in $\Delta$AUPRC, respectively. Additionally, we evaluate the generalization ability of UniMatch on the Meta-Mol benchmark, where it demonstrates outstanding performance (Section 4.3).

## 2 BACKGROUND

### 2.1 PROBLEM DEFINITION

The few-shot molecular property prediction problem, as defined by ADKF-IFT (Chen et al., 2023) and MHNfs (Schimunek et al., 2023), involves training models on a set of tasks $\{\mathcal{T}_\tau\}_{\tau=1}^{N_t}$ sampled from the training set $\mathcal{D}_{train}$ to improve generalization to new tasks. Each task $\mathcal{T}_\tau$ includes a support set $\mathcal{S}_\tau = \{(\mathbf{x}_{\tau,i}, y_{\tau,i})\}_{i=1}^{N_\tau^s}$ and a query set $\mathcal{Q}_\tau = \{(\mathbf{x}_{\tau,j}, y_{\tau,j})\}_{j=1}^{N_\tau^q}$, where $\mathbf{x}_{\tau,i} \in \mathbb{R}^d$ and $\mathbf{x}_{\tau,j} \in \mathbb{R}^d$ represent molecular features, and $y_{\tau,i}, y_{\tau,j} \in \{0, 1\}$ indicate the molecular properties or activities. The support set $\mathcal{S}_\tau$ provides a few labeled examples for task-specific adaptation, while the query set $\mathcal{Q}_\tau$ is utilized to evaluate the model's performance on unseen examples.

### 2.2 PRELIMINARIES

Graph neural networks (GNNs) are designed to handle graph-structured data (non-Euclidean data) by aggregating information from neighboring nodes to learn effective representations (Zhou et al., 2020). Models such as GCN (Kipf & Welling, 2017), GIN (Xu et al., 2019), and GAT (Veličković et al., 2018) are widely used for graph classification and other related applications. In a graph $\mathcal{G} = \{\mathcal{V}, \mathcal{E}\}$, $\mathcal{V}$ represents the set of nodes, and $\mathcal{E}$ the set of edges. $\mathbf{h}_v^{(0)}$ represents the initial features of node $v$, and $\mathbf{b}_{u,v}$ denotes the features of the edge $e_{u,v}$ between nodes $u$ and $v$. At the $l^{\text{th}}$ layer, the representation $\mathbf{h}_v^{(l)}$ of node $v$ is updated in GNNs as follows:

$$\mathbf{h}_v^{(l)} = \text{UPDATE}^{(l)}\left(\mathbf{h}_v^{(l-1)}, \text{AGGREGATE}^{(l)}\left(\left\{\left(\mathbf{h}_v^{(l-1)}, \mathbf{h}_u^{(l-1)}, \mathbf{b}_{v,u}\right) \mid u \in \mathcal{N}(v)\right\}\right)\right), \quad (1)$$

where $\mathcal{N}(v)$ is the set of neighboring nodes of $v$. The AGGREGATE function combines features from neighboring nodes, and the UPDATE function updates the node features for the next layer.

## 3 METHOD

This section presents the model architecture (Section 4.1) and the meta-learning strategy (Section 4.2) of Universal Matching Networks, which respectively achieve explicit hierarchical molecular matching and implicit task-level matching. Figure 2 illustrates an overview of UniMatch framework.

### 3.1 ARCHITECTURE: EXPLICIT HIERARCHICAL MOLECULAR MATCHING

We propose a novel architecture for explicit hierarchical molecular matching, which captures and aligns complex molecular structures across multiple levels (atomic, substructural, and molecular) via hierarchical pooling and matching, enabling fine-grained comparison and similarity assessment from local to global scales. It establishes the foundation for implicit task-level matching.

**Encoding Module.** Following the mainstream graph-based few-shot molecular property prediction approaches (Wang et al., 2021; Guo et al., 2021; Lv et al., 2024; Chen et al., 2023), we adopt the widely used GIN (Xu et al., 2019) as the backbone of our method. In GNNs, each layer aggregates local information from nodes and their neighboring hops. As the network depth increases, the model incrementally aggregates hierarchical information, progressing from individual nodes to substructures and ultimately capturing the entire molecule.

To capture molecular representations at different levels, we employ mean pooling to aggregate node representations at each layer of the GNN. For a given task $\tau$, we first obtain the node representations

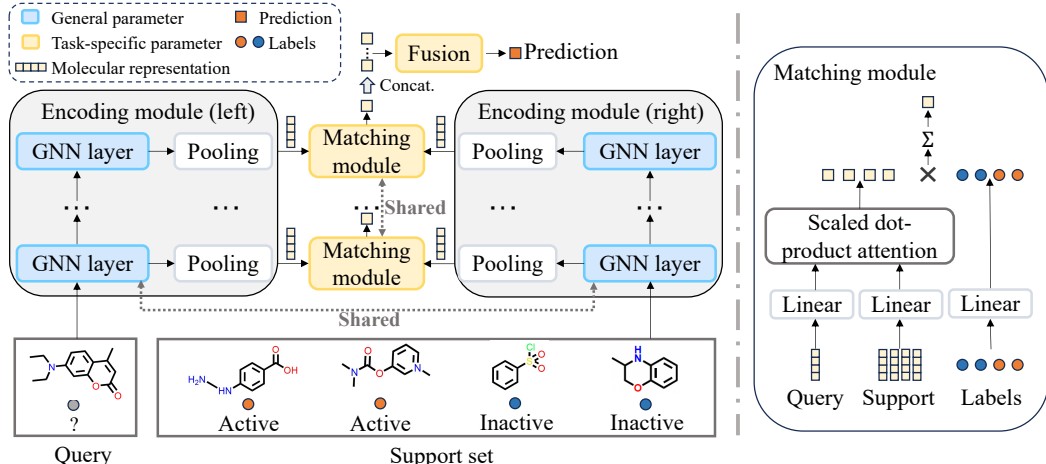

Figure 2: The overview of UniMatch. **Left:** Our model follows a hierarchical pooling-matching architecture comprising two components: an encoding module (including pooling) and a matching module. First, mean pooling is applied at each GNN layer to generate multi-level molecular representations. Then, an attention mechanism is utilized to align representations between the support set and query set across different levels. Finally, predictions from different GNN layers are integrated to obtain the final results. **Right:** The detailed process of the matching module.

$\mathbf{h}_{\tau,s}^{(l)}[1] \in \mathbb{R}^{n_s \times d}$ for the support set $\mathcal{S}_\tau$ and $\mathbf{h}_{\tau,q}^{(l)}[2] \in \mathbb{R}^{n_q \times d}$ for the query set $\mathcal{Q}_\tau$. Then, we utilize mean pooling to derive the molecular representations $\mathbf{z}_{\tau,s}^{(l)} \in \mathbb{R}^{N_\tau^s \times d}$ for the support set $\mathcal{S}_\tau$ and $\mathbf{z}_{\tau,q}^{(l)} \in \mathbb{R}^{N_\tau^q \times d}$ for the query set $\mathcal{Q}_\tau$, as follows:

$$\mathbf{z}_{\tau,s}^{(l)} = \text{Pooling}(\mathbf{h}_{\tau,s,v}^{(l)}, v \in \mathcal{V}_{\tau,s}), \mathbf{z}_{\tau,q}^{(l)} = \text{Pooling}(\mathbf{h}_{\tau,q,v}^{(l)} \ v \in \mathcal{V}_{\tau,q}), \tag{2}$$

where the Pooling function denotes mean pooling, and $l$ refers to the $l^{\text{th}}$ layer of the GNN.

**Matching Module.**  Hierarchical matching plays a critical role in our approach, as it allows the model to capture structural details at different levels—from atoms to entire molecules—ensuring a more accurate and fine-grained similarity identification. To achieve this, we adopt the attention mechanism introduced by Vaswani et al. (2017), which dynamically weighs the contributions of molecular features at multiple scales. Specifically, we designate the molecular representations $\mathbf{z}_{\tau,s}^{(l)} \in \mathbb{R}^{N_\tau^s \times d}$ in the support set $\mathcal{S}_\tau$ as the key, and the molecular representations $\mathbf{z}_{\tau,q}^{(l)} \in \mathbb{R}^{N_\tau^q \times d}$ in the query set $\mathcal{Q}_\tau$ as the query. The corresponding ground-truth labels $\mathbf{y}_{\tau,s} \in \mathbb{R}^{N_\tau^s \times 1}$ in the support set serve as the value. This attention-based approach allows for precise matching at the specified level:

$$\hat{\mathbf{y}}_{\tau,q}^{(l)} = \text{Softmax}(\frac{(\mathbf{z}_{\tau,q}^{(l)}\mathbf{W}_q)(\mathbf{z}_{\tau,s}^{(l)}\mathbf{W}_k)^\top}{\sqrt{d}})\mathbf{y}_{\tau,s}, \tag{3}$$

where $d$ is the dimension of molecular representations, and $\mathbf{W}_q, \mathbf{W}_k \in \mathbb{R}^{d \times d}$.

**Fusion.**  We repeat the steps above to obtain the matching results $\hat{\mathbf{y}}_{\tau,q}^{(l)}$ for each GNN layer. These results are concatenated into a joint representation, which is then passed through a linear layer to generate the final prediction $\hat{\mathbf{y}}_{\tau,q} \in \mathbb{R}^{N_\tau^q \times 2}$:

$$\hat{\mathbf{y}}_{\tau,q} = \text{Linear}_{\mathbf{W}_o}(\text{Concat}(\hat{\mathbf{y}}_{\tau,q}^{(1)}, \hat{\mathbf{y}}_{\tau,q}^{(2)}, \cdots, \hat{\mathbf{y}}_{\tau,q}^{(L)})), \tag{4}$$

where L represents the total number of layers in the GNN, and $\mathbf{W}_o \in \mathbb{R}^{L \times 2}$ are the parameters of the Linear function. For classification tasks, $\hat{\mathbf{y}}_{\tau,q}$ is normalized into a probability distribution using Softmax. By fusing multi-level features, the model captures the complex relationships within molecular structures, resulting in more robust and comprehensive predictions.

---

[1] the subscript s represents it belongs to support set.

[2] the subscript q represents it belongs to query set.

## 3.2 META-LEARNING

We employ a meta-learning strategy to facilitate implicit task-level matching. To further clarify why meta-learning functions as implicit task-level matching, we introduce a task relationship matrix that captures task similarities at a higher level of abstraction. This matrix allows for efficient adaptation and enhances generalization through optimized learning.

### 3.2.1 TRAINING AND INFERENCE

For simplicity, we define UniMatch as $f_{\boldsymbol{\theta},\mathbf{w}}$, where $\boldsymbol{\theta}$ denotes the parameters of the molecular encoder, and $\mathbf{w} = \{\mathbf{W}_q, \mathbf{W}_k, \mathbf{W}_o\}$ represents the parameters of the matching and fusion modules.

**Training Phase.** During the training phase, we employ a standard meta-learning process to improve the model's generalization ability using the training set $\mathcal{D}_{\text{train}}$, as illustrated in Algorithm 1. Parameter learning involves a combination of inner and outer optimization. To facilitate dual optimization, we split the support set $\mathcal{S}_{\text{train},\tau}$ for each task $\mathcal{T}_{\text{train},\tau}$ into $\mathcal{S}'_\tau$ and $\mathcal{Q}'_\tau$. The training loss $\mathcal{L}(\mathcal{Q}'_\tau, f_{\boldsymbol{\theta},\mathbf{w}})$ evaluated on $\mathcal{Q}'_\tau$ is defined as:

$$\mathcal{L}(\mathcal{Q}'_\tau, f_{\boldsymbol{\theta},\mathbf{w}}) = \sum_{(\mathbf{x}_{\tau,i}, y_{\tau,i}) \in \mathcal{Q}'_\tau} -\mathbf{y}_{\tau,i}^\top \cdot \log(\hat{\mathbf{y}}_{\tau,i}), \tag{5}$$

where $\mathbf{y}_{\tau,i} \in \mathbb{R}^2$ is a one-hot vector representing the class of the sample, where a positive sample is denoted by $[1, 0]$ and a negative sample is denoted by $[0, 1]$.

- **Inner Loop.** During the inner optimization, task-specific parameters $\mathbf{w}_\tau$ are updated for each task $\mathcal{T}_{\text{train},\tau}$, enabling the model to adapt quickly to the current task, as follows:

$$\mathbf{w}_\tau = \mathbf{w} - \alpha \nabla_{\mathbf{w}} \mathcal{L}(\mathcal{Q}'_\tau, f_{\boldsymbol{\theta},\mathbf{w}}), \tag{6}$$

where $\boldsymbol{\theta}$ denotes the fixed parameters in the inner optimization and $\alpha$ is the inner learning rate.

- **Outer Loop.** The outer optimization aims to update the meta-parameter ($\boldsymbol{\theta}$ and $\mathbf{w}$) to improve generalization across all tasks. During training, this is achieved by minimizing the aggregated loss over all tasks:

$$\boldsymbol{\theta}^*, \mathbf{w}^* = \arg\min_{\boldsymbol{\theta},\mathbf{w}} \sum_{\tau=1}^{N_t} \mathcal{L}(\mathcal{Q}_{\text{train},\tau}, f_{\boldsymbol{\theta},\mathbf{w}_\tau}), \tag{7}$$

where $N_t$ is the total number of training tasks.

**Inference Phase.** After training, we evaluate the model on a set of test tasks $\mathcal{T}_{\text{test}}$ drawn from the test set $\mathcal{D}_{\text{test}}$. The support set $\mathcal{S}_{\text{test},\tau}$ is split into $\mathcal{S}''_\tau$ and $\mathcal{Q}''_\tau$. With $\boldsymbol{\theta}$ fixed, we fine-tune the task-specific parameters $\mathbf{w}$ using $\mathcal{Q}''_\tau$ as defined in Eq. 6. After fine-tuning, the model is evaluated on $\mathcal{S}_{\text{test},\tau}$, which serves as the support set to predict the labels for the unknown query molecules.

### 3.2.2 IMPLICIT TASK-LEVEL MATCHING

In our UniMatch framework, we consider meta-learning as an implicit task-level matching mechanism. During the training phase, the model autonomously captures and internalizes inter-task relationships and features, embedding them into its meta-parameters. As a result, even without explicitly modeling task relationships, the model can rapidly adapt to new tasks, demonstrating a form of implicit matching.

Implicit matching primarily captures the model's adaptability across multiple tasks through latent relationships, rather than direct parameter updates. To quantify these relationships, we define a task relationship matrix $\mathbf{M} \in \mathbb{R}^{N_t \times N_t}$, where each element $\mathbf{M}_{i,j}$ represents the relationship (e.g., similarity or distance) between task $\mathcal{T}_{\text{train},i}$ and task $\mathcal{T}_{\text{train},j}$, which is defined as follows:

$$\mathbf{M}_{i,j} = g_{\boldsymbol{\theta},\mathbf{w}}(\mathcal{T}_{\text{train},i}, \mathcal{T}_{\text{train},j}), \tag{8}$$

where $g_{\boldsymbol{\theta},\mathbf{w}}$ is a relation function based on the meta-parameter ($\boldsymbol{\theta}$ and $\mathbf{w}$), measuring the relationship between task $\mathcal{T}_{\text{train},i}$ and task $\mathcal{T}_{\text{train},j}$.

- **Inner Loop.** We further extend the concept of implicit matching to directly shape the representation of task-specific adaptation parameters $\mathbf{w}_\tau$. In traditional explicit training, task parameters $\mathbf{w}_\tau$ are determined via gradient updates of the meta-parameter as shown in Eq. 6. By contrast, implicit matching enables a new formulation of these parameters as follows:

$$\mathbf{w}_\tau = \mathbf{w}_\tau + \sum_{j=1}^{N_t} \mathbf{M}_{\tau,j} \cdot (\mathbf{w}_j - \mathbf{w}_\tau). \tag{9}$$

This approach replaces explicit gradient updates with implicit parameter matching, aligning task-specific parameters to the optimal solution based on inter-task relationships.

- **Outer Loop.** After completing the inner-loop training, the implicit relationship matrix $\mathbf{M}$ is updated. Using the updated $\mathbf{M}$, the general parameter $\boldsymbol{\theta}$ is further refined. The update formula can be rewritten as:

$$\boldsymbol{\theta} = \boldsymbol{\theta} + \eta \sum_{i=1}^{N_t} \sum_{j=1}^{N_t} \mathbf{M}_{i,j} \cdot (\mathbf{w}_j - \mathbf{w}_i), \tag{10}$$

where $\eta$ is the outer learning rate. This formula uses $\mathbf{M}$ to guide parameter updates, emphasizing the core role of implicit matching in training phase. Simultaneously, the task-specific parameters $\mathbf{w}$ can be updated using Eq. 9.

Further details on the training and inference phase are provided in Appendix A.

## 4 EXPERIMENT

In this section, we evaluate the empirical performance of UniMatch, as outlined in Section 3. We validate our UniMatch on the MoleculeNet (Section 4.1) and FS-Mol (Section 4.2) benchmarks. Additionally, we perform an ablation study of UniMatch in Section 4.2. To demonstrate the generalization of UniMatch, we further test it on seven datasets from the Meta-MolNet benchmark in Section 4.3, covering both single-task and multi-task scenarios. Finally, we conduct visualization experiments in Section 4.4 to demonstrate the importance of the dual matching mechanism in UniMatch. Additional experiments and analyses are provided in Appendix E.3. All experiments are run on an NVIDIA RTX A6000 GPU.

### 4.1 FEW-SHOT MOLECULAR PROPERTY PREDICTION ON THE MOLECULENET BENCHMARK

Table 1: Comparison of all methods on the MoleculeNet benchmark with a support set size of 20. The mean test performance is reported as AUROC% along with the standard deviations.

| Method | Tox21 (12) ↑ | SIDER (27) ↑ | MUV (17) ↑ | ToxCast (617) ↑ |
|---|---|---|---|---|
| CHEF (Adler et al., 2020) | $61.97 \pm 0.65$ | $57.34 \pm 0.82$ | $53.17 \pm 4.21$ | $56.52 \pm 1.24$ |
| MixHop (Abu-El-Haija et al., 2019) | $78.14 \pm 0.33$ | $72.01 \pm 0.87$ | $78.04 \pm 3.01$ | $77.19 \pm 0.93$ |
| Siamese (Koch et al., 2015) | $80.40 \pm 0.35$ | $71.10 \pm 4.32$ | $59.59 \pm 5.13$ | - |
| ProtoNet (Snell et al., 2017) | $74.98 \pm 0.32$ | $64.54 \pm 0.89$ | $65.88 \pm 4.11$ | $63.70 \pm 1.26$ |
| MAML (Finn et al., 2017) | $80.21 \pm 0.24$ | $70.43 \pm 0.76$ | $63.90 \pm 2.28$ | $66.79 \pm 0.85$ |
| TPN (Liu et al., 2018) | $76.05 \pm 0.24$ | $67.84 \pm 0.95$ | $65.22 \pm 5.82$ | $62.74 \pm 1.45$ |
| EGNN (Kim et al., 2019) | $81.21 \pm 0.16$ | $72.87 \pm 0.73$ | $65.20 \pm 2.08$ | $63.65 \pm 1.57$ |
| IterRefLSTM (Altae-Tran et al., 2017) | $81.10 \pm 0.17$ | $69.63 \pm 0.31$ | $45.56 \pm 5.12$ | - |
| PAR (Wang et al., 2021) | $82.06 \pm 0.12$ | $\mathbf{74.68 \pm 0.31}$ | $66.48 \pm 2.12$ | $69.72 \pm 1.63$ |
| ADKF-IFT (Chen et al., 2023) | $82.43 \pm 0.60$ | $67.72 \pm 1.21$ | $\mathbf{98.18 \pm 3.05}$ | $72.07 \pm 0.81$ |
| MHNFs (Schimunek et al., 2023) | $80.23 \pm 0.84$ | $65.89 \pm 1.17$ | $73.81 \pm 2.53$ | $74.91 \pm 0.73$ |
| **UniMatch (Ours)** | $\mathbf{82.62 \pm 0.43}$ | $68.13 \pm 1.54$ | $79.40 \pm 3.14$ | $\mathbf{77.74 \pm 0.75}$ |
| Pre-GNN (Hu et al., 2020) | $82.14 \pm 0.08$ | $73.96 \pm 0.08$ | $67.14 \pm 1.58$ | $73.68 \pm 0.74$ |
| GNN-MAML (Guo et al., 2021) | $82.97 \pm 0.10$ | $75.43 \pm 0.21$ | $68.99 \pm 1.84$ | - |
| Pre-PAR (Wang et al., 2021) | $84.93 \pm 0.11$ | $78.08 \pm 0.16$ | $69.96 \pm 1.37$ | $75.12 \pm 0.84$ |
| Pre-ADKF-IFT (Chen et al., 2023) | $86.06 \pm 0.35$ | $70.95 \pm 0.60$ | $\mathbf{95.74 \pm 0.37}$ | $76.22 \pm 0.13$ |
| **Pre-UniMatch (Ours)** | $\mathbf{86.35 \pm 0.13}$ | $\mathbf{80.34 \pm 0.45}$ | $86.35 \pm 0.76$ | $\mathbf{81.63 \pm 0.73}$ |

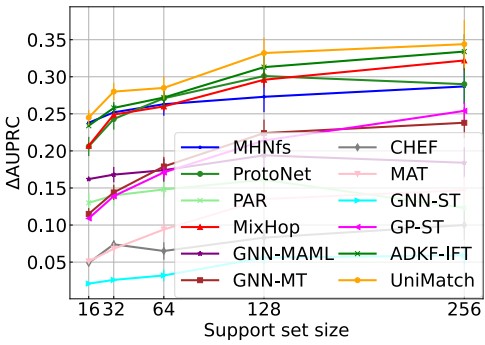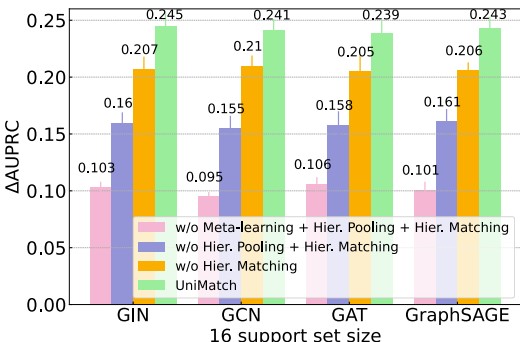

Figure 3: Mean performance with standard errors on the FS-Mol test tasks. (a) Performance of all compared approaches on the FS-Mol benchmark. (b) Ablation study of the dual matching mechanism in UniMatch across different backbones.

**Benchmark and Baselines.** MoleculeNet (Wu et al., 2018) serves as a benchmark for few-shot molecular property prediction, focusing on small molecules with molecular weights below 900 Daltons. This benchmark includes 4 datasets—Tox21, SIDER, MUV, and ToxCast—which contain 12, 27, 17, and 617 tasks, respectively. We compare **UniMatch** with two types of baselines: 1) Methods trained from scratch, including **CHEF** (Adler et al., 2020), **MixHop** (Abu-El-Haija et al., 2019), **Siamese** (Koch et al., 2015), **ProtoNet** (Snell et al., 2017), **MAML** (Ren et al., 2018), **TPN** (Liu et al., 2018), **EGNN** (Kim et al., 2019), **IterRefLSTM** (Altae-Tran et al., 2017), **PAR** (Wang et al., 2021), **MHNfs** (Schimunek et al., 2023), and **ADKF-IFT** (Chen et al., 2023); 2) Methods that fine-tune pretrained models, including **Pre-GNN** (Hu et al., 2020), **GNN-MAML** (Guo et al., 2021), **Pre-PAR** (Wang et al., 2021), and **Pre-ADKF-IFT** (Chen et al., 2023). **Pre-UniMatch** is **UniMatch** that utilizes pretrained parameters from the Pre-GNN model (Hu et al., 2020). More details are provided in Appendix B.

**Evaluation Procedure.** Following the procedural framework of Wang et al. (2021), we adopt AU-ROC (the area under the receiver operating characteristic curve) as the evaluation metric and set the support set size at 20 (i.e., 2-way 10-shot). The model is trained using the Adam optimizer (Kingma & Ba, 2014). During testing, results are averaged from 10 repeated experiments with different random seeds. For the baselines, we replicate the results for CHEF, MixHop, and MHNfs, while results for the other baselines are cited from Chen et al. (2023). Further details of the experimental setup can be found in Appendix B.2.

**Performance.** Table 1 demonstrates that both **UniMatch** and **Pre-UniMatch** outperform existing state-of-the-art methods on the Tox21, SIDER (pre-training stage only), and ToxCast datasets, achieving an average improvement of **2.87**%. Compared to CHEF, UniMatch demonstrates superior performance, suggesting that graph structures are more effective than fixed fingerprints for hierarchical representation learning in this context. Additionally, our UniMatch outperforms MixHop, highlighting the importance of hierarchical matching for molecular property prediction, especially in few-shot scenarios. On the MUV dataset, UniMatch ranks second among all baselines, possibly due to the severe distribution imbalance inherent in the MUV dataset.

### 4.2 FEW-SHOT MOLECULAR PROPERTY PREDICTION ON THE FS-MOL BENCHMARK

**Benchmark and Baselines.** FS-Mol, introduced by Stanley et al. (2021), serves as a benchmark for few-shot molecular property prediction. It comprises 5,120 tasks, partitioned into 4,938 for training, 40 for validation, 157 for testing, covering a total of 233,786 compounds (see Appendix C.1). To evaluate **UniMatch**, we compare it against four types of baselines: 1) Single-task methods: single-task GP with Tanimoto kernel (**GP-ST**) (Ralaivola et al., 2005), single-task GNN (**GNN-ST**) (Gilmer et al., 2017), **MixHop** (Abu-El-Haija et al., 2019), and **CHEF** (Adler et al., 2020); 2) Multi-task pre-training: Multi-task GNN (**GNN-MT**) (Stanley et al., 2021); 3) Self-supervised pre-training: Molecule Attention Transformer (**MAT**) (Maziarka et al., 2020); and 4) Meta-learning methods: **PAR** (Wang et al., 2021), **ProtoNet** (Snell et al., 2017), **GNN-MAML** (Guo et al., 2021),

**ADKF-IFT** (Chen et al., 2023), and **MHNfs** (Schimunek et al., 2023). Further details can be found in Appendix C.2.

**Evaluation Procedure.** We follow the experimental setup of the FS-Mol benchmark (Stanley et al., 2021). For each task, we employ unbalanced sampling to create an uneven distribution of positive and negative samples within the support set. The evaluation metric, $\Delta$AUPRC, is used to effectively assess the model's ability to improve minority class prediction, which is critical in imbalanced datasets (see Appendix C.3). During testing, we set five different sizes for the support set: 16, 32, 64, 128, and 256. For each size, we perform 10 repeated random splits of the support/query sets for the test tasks under these settings and take their averages as the final results.

**Performance.** Figure 3a displays the results of all compared methods. The results indicate that **UniMatch** outperforms all benchmarks across various support set sizes. It achieves substantial performance gains of **4.27**%, **8.53**%, **4.40**%, **6.07**%, and **4.26**% with support set sizes of 16, 32, 64, 128, and 256, respectively. These findings highlight the effectiveness of UniMatch's dual matching mechanism in enhancing generalization and robustness. Additionally, UniMatch demonstrates strong adaptability, consistently improving performance across different support set sizes.

**Ablation Study.** 1) To explore the importance of explicit hierarchical molecular matching (i.e., hierarchical pooling and matching) and implicit task-level matching (i.e., meta-learning) in capturing complex molecular structures, we use several common GNNs as baselines. UniMatch extends these baselines by incorporating these hierarchical mechanisms and a meta-learning strategy. 2) To evaluate the transferability of UniMatch, we test it on several common GNN architectures, including GIN (Xu et al., 2019), GCN (Kipf & Welling, 2017), GAT (Veličković et al., 2018), and Graph-SAGE (Hamilton et al., 2017). Experimental results, as shown in Figure 3b, highlight the significant advantages of the dual matching mechanism in effectively processing complex molecular structures and its strong adaptability and transferability across different GNNs.

**Sub-benchmark Performance.** The FS-Mol benchmark (157 test tasks) is divided into 7 subset tasks (Stanley et al., 2021). The results of UniMatch and baselines on these subset tasks are presented in Appendix E.2. Table E.2 shows the UniMatch outperforms state-of-the-art methods.

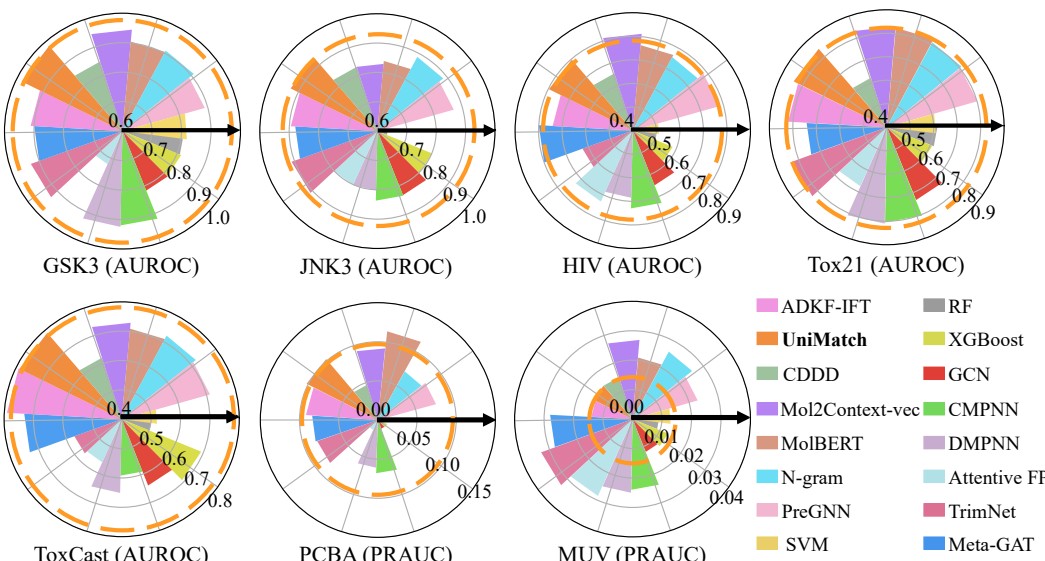

Figure 4: The performance of all compared methods on the seven classification tasks with a support set of size 2 on the Meta-MolNet benchmark. Each colored sector represents a method, with the height of the sector indicating its effectiveness. Starting from the black arrow, the methods are listed in the legend in a counterclockwise direction. **UniMatch** corresponds to the orange sector. The dashed orange circle marks the results of UniMatch. Methods with sectors below this line do not surpass UniMatch, while those above it show superior performance.

### 4.3 CROSS-DOMAIN DRUG DISCOVERY ON THE META-MOLNET BENCHMARK

**Benchmark and Baselines.** Meta-MolNet (Lv et al., 2024) sets a standard for evaluating generalization in computational chemistry by improving data quality and testing rigor. We evaluate our model on classification tasks including GSK3, JNK3, HIV, Tox21, ToxCast, PCBA, and MUV. For comparison, we consider four types of baselines: 1) Classical machine learning methods: support vector machine (**SVM**) (Bao et al., 2016), extreme gradient boosting (**XGBoost**) (Deng et al., 2021), and random forests (**RF**) (Fabris et al., 2018). 2) Supervised learning methods: **GCN** (Kipf & Welling, 2016), **CMPNN** (Song et al., 2020), **DMPNN** (Yang et al., 2019), **Attentive FP** (Xiong et al., 2019), and **TrimNet** (Li et al., 2020). 3) Self-supervised learning methods: **CDDD** (Winter et al., 2019), **Mol2Context-vec** (Lv et al., 2021), **MolBERT** (Fabian et al., 2020), **N-gram** (Liu et al., 2019), and **Pre-GNN** (Hu et al., 2020). 4) Meta-learning methods: **ADKF-IFT** (Chen et al., 2023) and **Meta-GAT** (Lv et al., 2024). All baseline results are reproduced according to Lv et al. (2024). Due to the sub-task settings of Meta-MolNet, prototype-based methods are no longer applicable. Further details can be found in Appendix D.1.

**Evaluation Procedure.** To evaluate the generalization ability of UniMatch, we follow a higher ratio of molecules/scaffolds by Lv et al. (2024). For classification tasks, we use AUROC and AUPRC as evaluation metrics. Specifically, AUROC is used to measure the performance of binary classification tasks (GSK3, JNK3, HIV, Tox21, and ToxCast), while AUPRC is more suitable for tasks with severely imbalanced distributions (PCBA, MUV). All experimental results are averaged over three independent runs with different random seeds, using a support set size of 2. Additional details on the evaluation metrics can be found in Appendix D.3.

**Performance.** Figure 4 presents a comparison of different methods across the seven classification datasets in Meta-MolNet benchmark. The results indicate that UniMatch performs exceptionally well on the GSK3, JNK3, Tox21, and ToxCast datasets, while showing less well on the HIV and PCBA datasets. Our method encounters significant challenges on the MUV dataset, likely due to distributional biases. Overall, UniMatch exhibits excellent generalization capabilities across most datasets for new molecular scaffolds, but struggles in specific cases, such as the MUV dataset.

### 4.4 VISUALIZATION

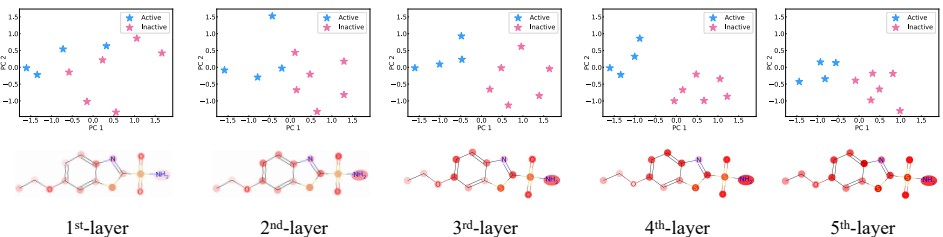

Figure 5: Layer-wise visualization for NR-AhR toxicity prediction. The first row presents PCA projections of 10 molecules, distinguishing active (blue) from inactive (pink) molecules. The second row displays an internal visualization of a selected molecule across different layers, where color intensity indicating shifts in the model's attention as the layers deepen.

To validate the importance of hierarchical representations, we visualize 10 molecules from the tox21 dataset for the NR-AhR toxicity prediction task, as shown in Figure 5. In the second row, we select one molecule (SMILES: "CCOc1ccc2nc(S(N)(=O)=O)sc2c1") to illustrate how each GNN layer captures distinct structural levels, ranging from atoms and substructures to the entire molecule. Additionally, PCA projections of these 10 molecules are used to examine the distribution of active and inactive molecules. This analysis enhances our understanding of the model's ability to distinguish molecular structures across layers, offering insights into the role of hierarchical feature extraction and its interpretability in toxicity prediction. Further details can be found in Appendix F.

## 5 RELATED WORK

### 5.1 GRAPH-BASED MOLECULAR PROPERTY PREDICTION

Graph-based methods are a mainstream approach for the few-shot molecular property prediction task. PAR (Wang et al., 2021) and ADKF-IFT (Chen et al., 2023) employ GIN (Xu et al., 2019) as the molecular encoder, while Meta-MGNN (Guo et al., 2021) utilizes Pre-GIN (Hu et al., 2020). Meta-GAT (Lv et al., 2024) adopts GAT (Veličković et al., 2018) to learn molecular representations. However, these methods typically focus on single-scale molecular features and overlook the hierarchical nature of molecular structures (Altae-Tran et al., 2017; Ren et al., 2018; Zhuang et al., 2023). In addition, several approaches (Zhao et al., 2023; Liu et al., 2024a) combine the strengths of Large Language Models (LLMs) to tackle the few-shot problem, but these methods often incur high computational costs. Our method differs by incorporating molecular hierarchical structures through hierarchical pooling and matching, enabling more effective alignment of complex structures.

### 5.2 MATCHING LEARNING

To address the few-shot learning problem, matching learning compares new instances with a small set of labeled examples to facilitate accurate predictions. Common methods include Matching Networks (Vinyals et al., 2016), ProtoNet (Snell et al., 2017), Relation Networks (Sung et al., 2018), and LGM-Net (Li et al., 2019). While these methods perform well in Natural Language Processing (NLP) and Computer Vision (CV), they struggle with the inherent complexity of molecular graphs, which feature non-Euclidean structures and intricate relationships between nodes and edges. Hierarchical matching can mitigate this issue by capturing multi-level representations, but existing approaches still face limitations when applied to molecular data due to its unique topological complexity. Specifically, AMN (Mai et al., 2019) and SSF-HRNet (Zhong et al., 2023), despite their improvements in feature robustness and hierarchical relationships, struggle to fully represent global structural information and generalize across complex, varied molecular graphs. Similarly, VTM (Kim et al., 2023) and HCL (Zheng et al., 2022) integrate hierarchical matching with patch-level techniques in CV, but their effectiveness diminishes when handling the structural diversity of molecular graphs. To overcome these challenges, our UniMatch combines explicit intra-molecular hierarchical learning with attention mechanisms at atomic, substructural, and molecular levels, along with implicit task-level hierarchical learning via meta-learning, enhancing the model's ability to capture task-specific molecular information and improve generalization.

## 6 CONCLUSION

We propose Universal Matching Networks (UniMatch) to address the limitations of existing few-shot learning methods in drug discovery. UniMatch employs a dual matching framework that integrates explicit molecular matching with implicit task-level matching. Explicit hierarchical molecular matching provides contextual representations that support implicit task-level matching, enabling better knowledge sharing across tasks. The complementary nature of these two mechanisms further enhances model performance and adaptability to new tasks. Experimental results show that UniMatch improves AUROC and $\Delta$AUPRC by 2.87% and 6.52%, respectively, on the MoleculeNet, and FS-Mol benchmarks and demonstrates excellent generalization on the Meta-MolNet benchmark. Future work will focus on improving the fusion mechanism of UniMatch by adopting advanced techniques such as attention fusion or multi-scale feature aggregation to better capture the complex relationships between structural levels. Additional details and discussions are provided in Appendix H.

**Acknowledgement.** This work was supported by the National Science and Technology Major Project 2023ZD0120802 and the National Key Research and Development Program of China 2022YFB4500300.

**Ethics Statement.** This paper addresses few-shot molecular representation learning without introducing new datasets or requiring human annotation. To the best of our knowledge, no additional ethical concerns arise beyond those commonly associated with research in this field.

**Reproducibility Statement.** We have detailed our model design in the paper, covering the architecture (explicit hierarchical molecular matching in Section 3.1) and meta-learning-based training and inference (implicit task-level matching in Section 3.2). Further training details are provided in the Appendix. We also outline the exact implementation of UniMatch in Section 4 and Appendix. Lastly, we will involve in the discussion regarding implementation details of our paper.

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

# A DETAILS OF TRAINING AND INFERENCE

## A.1 ALGORITHM

---

**Algorithm 1** Meta-training procedure for UniMatch.

---

**Input:** The few-shot training tasks $\{\mathcal{T}_\tau\}_{\tau=1}^{N_t}$ of molecular property prediction;
**Output:** Trained model $f_{\boldsymbol{\theta},\mathbf{w}}$;
1: Randomly initialize $\boldsymbol{\theta}$ and $\mathbf{w}$;
2: **while** not converged **do**
3:      Sample a batch $\mathcal{B}$ of tasks $\{\mathcal{T}_\tau | \tau \in \mathcal{B}\}$;
4:      **for** all $\{\mathcal{T}_\tau | \tau \in \mathcal{B}\}$ **do**
5:          Sample $N_\tau^s$ and $N_\tau^q$ molecules to form $\mathcal{S}_{\text{train},\tau}$ and $\mathcal{Q}_{\text{train},\tau}$;
6:          Split $\mathcal{S}_{\text{train},\tau}$ into $\mathcal{S}'_\tau$ and $\mathcal{Q}'_\tau$;
7:          **for** $l = 1, \ldots, L$ **do**
8:              Obtain node representations $\mathbf{h}_{\tau,s'}^{(l)}$, $\mathbf{h}_{\tau,q'}^{(l)}$, and $\mathbf{h}_{\tau,q}^{(l)}$ of $l^{\text{th}}$ GNN layer by Eq. 1;
9:              Obtain molecular representations $\mathbf{z}_{\tau,s'}^{(l)}$, $\mathbf{z}_{\tau,q'}^{(l)}$, and $\mathbf{z}_{\tau,q}^{(l)}$ of $l^{\text{th}}$ GNN layer by Eq. 2;
10:             Evaluate prediction $\hat{\mathbf{y}}_{\tau,q'}^{(l)}$ and $\hat{\mathbf{y}}_{\tau,q}^{(l)}$ of $l^{\text{th}}$ GNN layer by Eq. 3;
11:          **end for**
12:          Evaluate the final prediction $\hat{\mathbf{y}}_{\tau,q'}$ and $\hat{\mathbf{y}}_{\tau,q}$ by Eq. 4;
13:          Evaluate training loss $\mathcal{L}\left(\mathcal{Q}'_\tau, f_{\boldsymbol{\theta},\mathbf{w}}\right)$ by Eq. 5;
14:          Fine-tune $\boldsymbol{\theta}$ as $\boldsymbol{\theta}_\tau$ by Eq. 6;
15:          Evaluate testing loss $\mathcal{L}\left(\mathcal{Q}_{\text{train},\tau}, f_{\theta,\mathbf{w}_\tau}\right)$ by Eq. 5;
16:      **end for**
17:      Update $\boldsymbol{\theta}$, $\mathbf{w}$ by Eq. 7;
18: **end while**

---

Algorithm 1 outlines the meta-training procedure for **UniMatch**, designed to optimize the model for few-shot molecular property prediction tasks. The algorithm starts by initializing the parameters $\boldsymbol{\theta}$ and $\mathbf{w}$ and iteratively updates them through a combination of inner and outer optimization steps. Within each iteration, a batch of tasks $\mathcal{T}_\tau$ is sampled, and the support set $S_{\text{train},\tau}$ is split into $S'_\tau$ and $Q'_\tau$. For each GNN layer, node and molecular representations are computed, and predictions are evaluated using the specified loss functions. The meta-parameters are fine-tuned and updated based on the performance on the query sets, ensuring effective model generalization across tasks.

## A.2 DETAILS OF IMPLICIT TASK-LEVEL MATCHING

**Training Phase.**

- **Task Vector $\mathbf{p}_\tau$ Expansion and Analysis**. Each task $\mathcal{T}_{\text{train},\tau}$ has its own internal task vector $\mathbf{p}_\tau$, which can be represented as an expansion derived from the general parameter $\boldsymbol{\theta}$, the task-specific parameter $\mathbf{w}$, and their relationships with other tasks. The relationship matrix $\mathbf{M}$ governs parameter sharing and task matching.

- **Using $\mathbf{p}_\tau$ to Construct Task Relationship Matrix $\mathbf{M}$**. We want to use the task vector $\mathbf{p}_\tau$ to construct the task relationship matrix $\mathbf{M}$. The following methods can be used to measure the similarity or relationship between task vectors:

  - **Dot Product Similarity**:

  $$\mathbf{M}_{\tau,j} = \mathbf{p}_\tau^\top \mathbf{p}_j. \tag{11}$$

  The dot product similarity measures the inner product of task vectors $\mathbf{p}_\tau$ and $\mathbf{p}_j$ in vector space. The larger the value, the higher the similarity between the two tasks.

  - **Cosine Similarity**:

  $$\mathbf{M}_{\tau,j} = \frac{\mathbf{p}_\tau^\top \mathbf{p}_j}{\|\mathbf{p}_\tau\|\|\mathbf{p}_j\|}. \tag{12}$$

  The cosine similarity measures the cosine of the angle between task vectors $\mathbf{p}_\tau$ and $\mathbf{p}_j$. The value is in the range of $[-1, 1]$, and the closer it is to 1, the more similar the two tasks are.

– **Euclidean Distance**:

$$\mathbf{M}_{\tau,j} = -\|\mathbf{p}_\tau - \mathbf{p}_j\|^2. \tag{13}$$

The Euclidean distance measures the difference between two task vectors. The smaller the value, the closer the relationship between the two tasks.

- **Using Relationship Matrix M for Task Matching**. Using the above similarity metrics, we can obtain a task relationship matrix $\mathbf{M}$, where each element $\mathbf{M}_{\tau,j}$ represents the similarity or matching degree between task $\mathcal{T}_{\text{train},\tau}$ and task $\mathcal{T}_{\text{train},j}$. Based on this relationship matrix, we can introduce inter-task information sharing in the inner loop and outer loop optimization.

- **Inner Loop Optimization Using Task Vector $\mathbf{w}_\tau$**. During the inner loop optimization, we can use the relationship matrix $\mathbf{M}$ to update the task-specific parameters $\mathbf{w}_\tau$ of task $\mathcal{T}_{\text{train},\tau}$. The inner update formula is as defined in Eq. 9.

- **Outer Loop Optimization Using Task Vector $\mathbf{p}_\tau$**. After $\mathbf{w}_\tau$ updated, task vectors $\mathbf{p}$ can be further refined, which in turn can be used to update the relationship matrix $\mathbf{M}$. During the outer loop optimization, we use the task relationship matrix $\mathbf{M}$ to update the general parameter $\boldsymbol{\theta}$, so that the model's performance on all tasks can be improved. The outer update formula is as defined in Eq. 10.

- **Relationship Matrix M Update**. Next, the task vectors $\mathbf{p}$ are further refined, and the relationship matrix $\mathbf{M}$ is updated based on the differences between these task vectors.

**Inference Phase.**  In the traditional inference phase, models typically employ explicit gradient descent strategies to update the parameters for a new task. This explicit update process relies on the new task's loss value and gradient calculations. However, under the framework of implicit task matching, we aim to directly generate adaptive parameters for the new task based on its matching relationship with training tasks, thereby avoiding explicit gradient update processes.

Assuming that the current model needs to handle a new task $\mathcal{T}_{\text{test},\tau}$, its adaptive parameters can be represented using the following implicit matching formula:

$$\mathbf{w}_j = \mathbf{w} + \sum_{k=1}^{N_{\text{test}}} \mathbf{M}_{j,k} \cdot (\mathbf{w}_k - \mathbf{w}_j), \tag{14}$$

where $N_{\text{test}}$ denotes the total number of test tasks.

Then, direct inference and prediction are performed using the fine-tuned new task parameters $\mathbf{w}_j$, thereby improving the sample prediction accuracy for the new task.

## B    DETAILS OF MOLECULENET BENCHMARK

In this section, we introduce the details of datasets that are included in the MoleculeNet benchmark in Section B.1. In addition, we show the details of the experimental setup B.2.

### B.1    DETAILS OF DATASETS

Table 2: Summary of datasets included in MoleculeNet.

| Dataset | Tox21 | SIDER | MUV | ToxCast |
|---|---|---|---|---|
| Compounds | 8,014 | 1,427 | 93,127 | 8,615 |
| Tasks | 12 | 27 | 17 | 617 |
| Meta-Training Tasks | 9 | 21 | 12 | 450 |
| Meta-Testing Tasks | 3 | 6 | 5 | 167 |

In the MoleculeNet benchmark, we perform experiments on 4 datasets in Tabel 2, which include Tox21 (Richard et al., 2020), SIDER (Kuhn et al., 2016), MUV (Rohrer & Baumann, 2009), and ToxCast (Richard et al., 2016). Widely utilized in the assessment of compound toxicity for drug development and environmental risk evaluation, the Tox21 dataset, as described in Richard et al.

(2020), contains 8,014 compounds categorized into 12 tasks. By analyzing this dataset, researchers can identify environmental pollutants and potential drug candidates, offering crucial insights into their impact on human health. The SIDER dataset, introduced in Kuhn et al. (2016) , serves as a crucial database of drug side effects, encompassing extensive information on medications and their associated adverse responses. This dataset encompasses 1427 compounds distributed among 27 categories. Utilizing the SIDER dataset provides researchers with valuable insights into drug safety profiles and potential side effects. The MUV dataset (Rohrer & Baumann, 2009), which includes 93,127 compounds distributed among 17 tasks showcasing a range of biological activities, is widely acknowledged as a key standard for evaluating the multifaceted functions of drug compounds. A fundamental resource in toxicology research, the ToxCast dataset (Richard et al., 2016) is a critical high-throughput screening database used to evaluate the potential health hazards posed by various compounds. With a compilation of 8,615 compounds and 617 tasks, this dataset significantly contributes to the field of toxicology.

## B.2 DETAILS OF EXPERIMENTAL SETUP

In HieMatch (and Pre-HirMatch), GIN used in Eq. 1 which consists of 5 layers with hidden size 300. In addition, attention mechanism used in Eq. 3 consist of 1 layer with 1 head. We implement Uni-Match in PyTorch (Paszke et al., 2019) and Pytorch Geometric library (Fey & Lenssen, 2019). We train the model for a maximum number of 5000 epoches. We employ the Adam optimizer (Kingma & Ba, 2014) with a learning rate of 0.001 for meta-learning, while using a higher learning rate of 0.05 for fine-tuning the matching module and fusion module within each task. The dropout rate is maintained at 0.1 for all components, except for the graph-based molecular encoder. We summarize the hyperparameters used by UniMatch in Table 3.

Table 3: Hyperparameters used by UniMatch

| Hyperparameter | Explored values | Selected |
|---|---|---|
| learning rate for meta-learning | 0.001 | 0.001 |
| learning rate for fine-tuning | 0.01∼0.5 | 0.05 |
| number of update steps for fine-tuning | 1∼5 | 5 |
| number of layer of GNN in Eq.1 | 5 | 5 |
| number of layer of matching module in Eq.3 | 1 | 1 |
| number of head of matching module in Eq.3 | 1 | 1 |
| dropout | 0.0∼0.5 | 0.1 |
| hidden dimension for GNN in Eq.1 | 300 | 300 |

## C DETAILS OF FS-MOL BENCHMARK

In this section, we first introduce the details of FS-Mol benchmark (Stanley et al., 2021) in Section C.1. The subsequent discussion delves into the details of the compared baselines on FS-Mol benchmark in Section C.1. In addition, further details regarding the evaluation metric $\Delta$AUPRC is presented in Section C.3. Finally, the details of experimental setup on FS-Mol benchmark is presented in Section C.4.

## C.1 DETAILS OF BENCHMARKS

The Few-Shot Learning Dataset of Molecules (**FS-Mol**) (Stanley et al., 2021) is designed for machine learning applications in the Quantitative Structure-Activity Relationships (QSAR) field (Tropsha et al., 2023), specifically focusing on few-shot learning scenarios. It comprises a total of 5120 distinct assays, encompassing 233,786 unique compounds. The dataset is partitioned into three subsets: $D_{train}$ for training, $D_{test}$ for testing, and $D_{valid}$ for validation purposes. $D_{test}$ contains 157 tasks, $D_{train}$ includes 4938 tasks, and $D_{valid}$ is composed of 40 tasks. Notably, each task in the dataset contains an average of 94 compounds, a notably lower figure compared to other similar datasets. This characteristic reflects the high specificity of the protein targets and the corresponding assays, posing a significant challenge in the QSAR domain.

## C.2 DETAILS OF BASELINES.

In the comparative analysis of the FS-Mol benchmark (Stanley et al., 2021), four types of baselines have been chosen: Single-task methods, Multi-task pre-training methods, Self-supervised pre-training methods, and Meta-learning methods.

**Single-task Methods.** The single-task methods are single-task GP with Tanimoto kernel (**GP-ST**) (Ralaivola et al., 2005), single-task GNN (**GNN-ST**) (Gilmer et al., 2017), **MixHop** (Abu-El-Haija et al., 2019) and **CHEF** (Adler et al., 2020) for context-enriched information.

**GP-ST**, as delineated in the study by (Ralaivola et al., 2005), encompassing the random walk kernel, shortest-path kernel, and subtree kernel, are employed to evaluate the resemblance between graphs of chemical compounds. Gilmer et al. (2017) introduces **GNN-ST**, particularly focusing on MPNNs for proficient learning from graph-based representations of molecules in quantum chemistry. **CHEF** (Adler et al., 2020) leverages fingerprint-based features to capture chemical information, thereby enhancing the performance of molecular property prediction tasks. **MixHop** (Abu-El-Haija et al., 2019) is a novel graph convolutional architecture that enables higher-order neighborhood mixing in Graph Neural Networks (GNNs). By incorporating multiple neighborhood feature mixing operations, including neighborhood difference operators, the MixHop model can learn a broader range of graph structural representations without increasing computational complexity.

**Multi-task Pre-training Method.** Multi-task GNN (**GNN-MT**) (Stanley et al., 2021) employs a 10-layer pre-trained GNN with 128 hidden dimensions and "principal neighborhood message aggregation." Task-specific readout functions and an MLP with a 512-dimensional hidden layer produce activity label predictions. The model is fine-tuned on all tasks in $\mathcal{D}_{train}$ using multi-task learning.

**Self-supervised Pre-training Method.** The Molecule Attention Transformer (**MAT**) (Maziarka et al., 2020) modifies the Transformer architecture (Vaswani et al., 2017) by incorporating insights on inter-atomic distances and the molecular graph structure into the self-attention mechanism.

**Meta-learning Methods.** Property-Aware Relation Networks (**PAR**) (Wang et al., 2021), Prototypical Networks (**ProtoNet**) (Snell et al., 2017), **GNN-MAML** (Guo et al., 2021), and **ADKF-IFT** (Chen et al., 2023) are four typical meta-learning methods. Specifically, **PAR** (Wang et al., 2021), introduces a property-aware embedding function that transforms generic molecular embeddings into a substructure-aware representation which relevant to the target property, and designs an adaptive relation graph learning module to jointly estimate the molecular relation graph and refine the molecular embeddings with respect to the target property. Schimunek et al. (2023) proposes **MHNfs** approach, utilizing a Modern Hopfield Network (MHN) (Ramsauer et al., 2020) to link molecules with an extensive array of reference molecules, thereby enhancing the covariance structure of the data and mitigating spurious correlations of molecules. **ProtoNet** (Snell et al., 2017), a simple approach to few-shot classification, learns an embedding where each class is represented by a prototype, computed as the mean of the embedded support examples for that class. Classification is then done by computing distances from the query example to each class prototype. **GNN-MAML** (Guo et al., 2021) uses graph neural networks to learn molecular representations, and employs a meta-learning framework for model optimization. It also incorporates molecular structure, self-supervised modules, and self-attentive task weights to exploit unlabeled data and address task heterogeneity. **ADKF-IFT** (Chen et al., 2023) combines the representational power of deep learning with the probabilistic modeling capabilities of gaussian processes, enabling efficient and uncertainty-aware molecular property prediction through meta-learning.

## C.3 EVALUATION METRICS OF FS-MOL BENCHMARK

The $\Delta$AUPRC (Area Under the Curve for Precision-Recall) serves as a pivotal statistical measure utilized for assessing enhancements in the efficacy of classification models when confronted with imbalanced datasets due to targeted modifications, like algorithmic adjustments or alterations in data processing methodologies. By contrasting the precision-recall curve's area prior to and post adjustments, this metric adeptly elucidates the extent of enhancement in the capacity of model to identify minority classes, thereby supplying a quantitative foundation for optimizing the model and facilitating decision-making support.

In line with the research conducted by Stanley et al. (2021), we employ the $\triangle$AUPRC as an evaluation metric for comparing all baseline models. The specific calculation formula is detailed below:

$$\triangle \mathrm{AUPRC}\left(f_{\boldsymbol{\theta},\mathbf{w}}\right) = \mathrm{AUPRC}\left(f_{\boldsymbol{\theta},\mathbf{w}}\right) - \frac{N_{\tau}^{q}(1)}{N_{\tau}^{q}}, \tag{15}$$

where the $N_{\tau}^{q}(1)$ represents the number of active molecules in query set $\mathcal{Q}_{\tau}$.

## C.4 DETAILS OF EXPERIMENTAL SETUP

In UniMatch, the hyperparameters used by UniMatch are reported in Table 3. What is more, on FS-Mol benchmark (Stanley et al., 2021), we set the batch task 21 and weight decay 5e-5. And we train the model for 10,000 epoches.

## D DETAILS OF META-MOLNET BENCHMARK

In this section, we first introduce the details of Meta-MolNet benchmark (Lv et al., 2024) in Section D.1. In addition, we provide the details of the baselines in Section D.2. Finally, the details of evaluation metric is provided in Section D.3.

## D.1 DETAILS OF BENCHMARKS

Meta-MolNet is an innovative benchmarking platform designed to improve molecular machine learning models by integrating diverse datasets through multitask and transfer learning, spanning applications from drug discovery to materials science. In this paper, we use 7 classification tasks on Meta-MolNet benchmark to evaluate our UniMatch, which include GSK3, JNK3, HIV, Tox21, Tox-Cast, PCBA and MUV. The GSK3 dataset focuses on predicting the activity of compounds against the GSK3 enzyme, which is associated with diseases like diabetes and Alzheimer's. The JNK3 dataset assesses the inhibitory activity of compounds against JNK3, a kinase implicated in neurodegenerative diseases. The HIV dataset contains data for predicting the ability of compounds to inhibit HIV replication. Tox21 evaluates the toxicity of compounds across multiple biological pathways, while ToxCast predicts the toxic effects of environmental chemicals. The PCBA dataset measures compound activity across various bioassays from the PubChem database. Lastly, the MUV dataset provides a rigorous and unbiased benchmark for validating virtual screening methods. Together, these tasks offer a comprehensive evaluation framework for molecular machine learning models. The detailed description of datasets in Table 4.

Table 4: Detailed Description of the benchmark datasets

| Task type | Datasets | Category | Data type | Tasks | No. of Molecules | No. of Scaffolds | Molecules/ Scaffolds ratio | Metrics | Threshold |
|---|---|---|---|---|---|---|---|---|---|
| Single Task Classification | GSK3 | Biophysics | SMILES | 1 | 3,197 | 38 | 84.13 | ROC-AUC | 30 |
| | JNK3 | Biophysics | SMILES | 1 | 4,873 | 62 | 78.60 | ROC-AUC | 30 |
| | HIV | Biophysics | SMILES | 1 | 6,386 | 68 | 93.91 | ROC-AUC | 30 |
| Multi Task Classification | Tox21 | Physiology | SMILES | 12 | 2,119 | 12 | 176.58 | ROC-AUC | 30 |
| | ToxCast | Physiology | SMILES | 617 | 2,372 | 14 | 169.43 | ROC-AUC | 30 |
| | PCBA | Biophysics | SMILES | 128 | 21,835 | 34 | 642.21 | PRC-AUC | 200 |
| | MUV | Biophysics | SMILES | 17 | 11,671 | 152 | 76.78 | PRC-AUC | 30 |

## D.2 DETAILS OF BASELINES

Four types of baselines—classical machine learning models, graph-based models, message passing neural networks, and self-supervised pre-training models—are chosen for comparative analysis on the Meta-MolNet benchmark (Lv et al., 2024).

**Classical Machine Learning Methods.** Support Vector Machines (**SVM**) (Bao et al., 2016), extreme gradient boosting algorithms (**XGBoost**) (Deng et al., 2021), and Random Forests (**RF**) (Fabris et al., 2018) are among the classical machine learning methods that utilize descriptors and/or fingerprints commonly found in traditional QSPR/QSAR models (Cherkasov et al., 2014). Notably,

the Extended Connectivity Fingerprints (ECFPs) (Rogers & Hahn, 2010; Glen et al., 2006) and Molecular ACCess System (MACCS) keys (Bender et al., 2004; Unterthiner et al., 2014) are widely used as fingerprints in such models. **SVM** (Bao et al., 2016) is a robust machine learning algorithm designed to identify the optimal solution for classification tasks by determining the maximum margin hyperplane within a high-dimensional space. **XGBoost** (Deng et al., 2021) is a proficient machine learning technique that utilizes distributed gradient boosting to provide rapid, adaptable, and user-friendly solutions. Information about **RF** (Fabris et al., 2018) can be found in Appendix C.1.

**Supervised Learning Methods.** Graph Convolutional Networks (**GCN**) (Duvenaud et al., 2015), Directed Message Passing Neural Networks (**DMPNN**) (Yang et al., 2019), Communicative Message Passing Neural Networks (**CMPNN**) (Song et al., 2020), **Attentive FP** (Xiong et al., 2019), and Triplet Message Networks (**TrimNet**) (Li et al., 2020) are among the supervised learning methods. Specifically, **GCN**[3] (Duvenaud et al., 2015) employs convolution operations based on the eigen decomposition of the Laplacian matrix, which allows them to aggregate information from neighboring nodes and derive node embedding representations. **DMPNN**[4] (Yang et al., 2019) use Laplacian eigen decomposition for convolution operations, aggregating information from neighboring nodes to derive node embeddings. **CMPNN**[5] (Song et al., 2020) enhance modeling of molecular properties by using a node-edge interaction module to effectively integrate atom and bond features. **Attentive FP**[6] (Xiong et al., 2019) employs atom and bond attributes to create feature vectors, preserving spatial information and capturing both local and nonlocal effects with a graph attention mechanism. **TrimNet**[7] (Li et al., 2020) utilizes a triplet message mechanism to extract edge information from atom-bond-atom interactions, achieving state-of-the-art performance.

**Self-supervised Learning Methods.** **CDDD** (Winter et al., 2019), **Mol2Context-vec** (Lv et al., 2021), **MolBERT**, **N-gram**, and **Pre-GNN** (Hu et al., 2020) are self-supervised methods that pre-train on large molecular datasets to extract meaningful descriptors. These data-driven approaches produce generalizable features, avoiding fixed extraction rules and reducing overfitting. Specifically, **CDDD**[8] (Winter et al., 2019) learns features from a large chemical structure corpus by translating between different molecular representations, compressing shared information into a low-dimensional vector. **Mol2Context-vec**[9] (Lv et al., 2021) uses a Bi-LSTM to create dynamic representations of molecular substructures, capturing intramolecular hydrogen bonds and other non-covalent interactions. **MolBERT**[10] (Fabian et al., 2020) is a Transformer-based model that uses BERT (Devlin et al., 2018) to learn high-quality molecular representations for drug discovery. **N-gram**[11] (Liu et al., 2019) captures co-occurrence patterns of local substructures by extracting n-grams from the graph and creating a histogram to represent their frequencies, forming the graph-level representation. **Pre-GNN**[12] (Hu et al., 2020) pre-trains graph neural networks by learning representations at both node and graph levels, capturing local and global structural information in molecular graphs.

**Meta-learning Methods.** **Meta-GAT** (Lv et al., 2024) and **ADKF-IFT** (Chen et al., 2023) are two typical meta-learning methods. Specifically, **Meta-GAT**[13] (Lv et al., 2024) s a graph attention network that uses cross-domain meta-learning to predict molecular properties with few examples. By extracting meta-knowledge from similar molecules across domains, it reduces sample complexity and quickly adapts to new scaffold molecules with minimal data. **ADKF-IFT**[14] (Chen et al., 2023) can be seen in Appendix C.1.

---

[3] https://github.com/tkipf/gcn.git
[4] https://github.com/chemprop/chemprop.git
[5] https://github.com/SY575/CMPNN.git
[6] https://github.com/OpenDrugAI/AttentiveFP.git
[7] https://github.com/yvquanli/trimnet.git
[8] https://github.com/jrwnter/cddd.git
[9] https://github.com/lol88/Mol2Context-vec.git
[10] https://github.com/BenevolentAI/MolBERT.git
[11] https://github.com/chao1224/n_gram_graph.git
[12] https://github.com/snap-stanford/pretrain-gnns.git
[13] https://github.com/lol88/Meta-MolNet.git
[14] https://github.com/Wenlin-Chen/ADKF-IFT.git

## D.3 EVALUATION METRICS OF META-MOLNET

In this paper, we use benchmark datasets with a higher ratio of molecules to scaffolds, presenting a significantly more challenging scenario compared to random cross-validation and datasets with a lower ratio (Lv et al., 2024), for evaluating generalization ability. For classification tasks, we use Area Under the Receiver Operating Characteristic Curve (**AUROC**) and Area Under the Precision-Recall Curve (**AUPRC**) as evaluation metrics. Specifically, **AUROC** measures the trade-off between the true positive rate (sensitivity) and the false positive rate (1 - specificity) across different classification thresholds. **AUROC** ranges from 0 to 1, where 0.5 represents a random classifier and 1 represents a perfect classifier. A higher **AUROC** value indicates better classification performance, making it well-suited for evaluating binary classification tasks such as GSK3, JNK3, HIV, Tox21, and ToxCast. Meanwhile, **AUPRC** considers the trade-off between precision (positive predictive value) and recall (sensitivity). Like **AUROC**, **AUPRC** ranges from 0 to 1, with higher values indicating better performance. **AUPRC** is particularly useful for evaluating models on imbalanced datasets, making it more suitable for tasks such as PCBA and MUV, which have severely skewed distributions.

## D.4 DETAILS OF EXPERIMENTAL SETUP

On the Meta-MolNet benchmark, we set the query set size to 8 and the support set size to 2. We employ the AdamW optimizer (Loshchilov & Hutter, 2017) with a learning rate of 0.001 for meta-learning and an inner learning rate of 0.001 for fine-tuning the task-specific modules within each task. A weight decay of 5e-4 is applied. The model is trained for 100 epochs to ensure robust performance.

# E FURTHER EXPERIMENTS RESULTS ON FS-MOL

## E.1 OVERALL PERFORMANCE

Figure 6 (a)~(e) show the performance of different methods in classifying 157 FS-Mol (Stanley et al., 2021) test tasks across various support set sizes via box plots. The box plots show the distribution of classification accuracies for each method, providing insight into their overall performance and effectiveness in handling varying support set sizes. Our HieMatch demonstrates superior performance compared to the state-of-the-art (SOTA) method across all metrics.

## E.2 SUB-BENCHMARK PERFORMANCE

FS-Mol (Stanley et al., 2021) divides tasks into 7 sub-benchmarks using Enzyme Commission (EC) numbers (Hu et al., 2012), allowing for assessment across the entire benchmark. In classification tasks with a support set size of 16, Table E.2 illustrates the performance of the top methods across all sub-benchmarks. The results highlight that, while excelling in overall performance, HieMatch emerges as the top performer in half of the sub-benchmarks for classification tasks.

Table 5: The classification performance for the 16 support set size.

| FS-Mol sub-benchmark (EC category) | | | Method | | | | |
|---|---|---|---|---|---|---|---|
| Class | Description | #tasks | RF | GP-ST | GNN-MAML | ADKF-IFT | UniMatch |
| 1 | oxidoreductases | 7 | $0.081 \pm 0.032$ | $0.013 \pm 0.019$ | $0.046 \pm 0.023$ | $0.103 \pm 0.0036$ | $\mathbf{0.231 \pm 0.075}$ |
| 2 | kinases | 125 | $0.082 \pm 0.006$ | $0.013 \pm 0.004$ | $0.178 \pm 0.009$ | $0.247 \pm 0.010$ | $\mathbf{0.256 \pm 0.012}$ |
| 3 | hydrolases | 20 | $0.158 \pm 0.026$ | $0.062 \pm 0.019$ | $0.106 \pm 0.024$ | $0.213 \pm 0.029$ | $0.201 \pm 0.028$ |
| 4 | lysases | 2 | $0.218 \pm 0.172$ | $0.161 \pm 0.112$ | $0.218 \pm 0.147$ | $0.223 \pm 0.160$ | $0.211 \pm 0.061$ |
| 5 | isomerases | 1 | $0.119 \pm 0.029$ | $-0.014 \pm 0.015$ | $0.006 \pm 0.021$ | $0.121 \pm 0.049$ | $0.087 \pm 0.025$ |
| 6 | ligases | 1 | $0.027 \pm 0.069$ | $-0.011 \pm 0.003$ | $0.001 \pm 0.017$ | $0.103 \pm 0.066$ | $\mathbf{0.359 \pm 0.011}$ |
| 7 | translocases | 1 | $0.102 \pm 0.053$ | $0.067 \pm 0.050$ | $0.001 \pm 0.021$ | $0.082 \pm 0.049$ | $-0.009 \pm 0.011$ |
| | all enzymes | 157 | $0.093 \pm 0.007$ | $0.021 \pm 0.005$ | $0.162 \pm 0.009$ | $0.230 \pm 0.009$ | $\mathbf{0.245 \pm 0.011}$ |

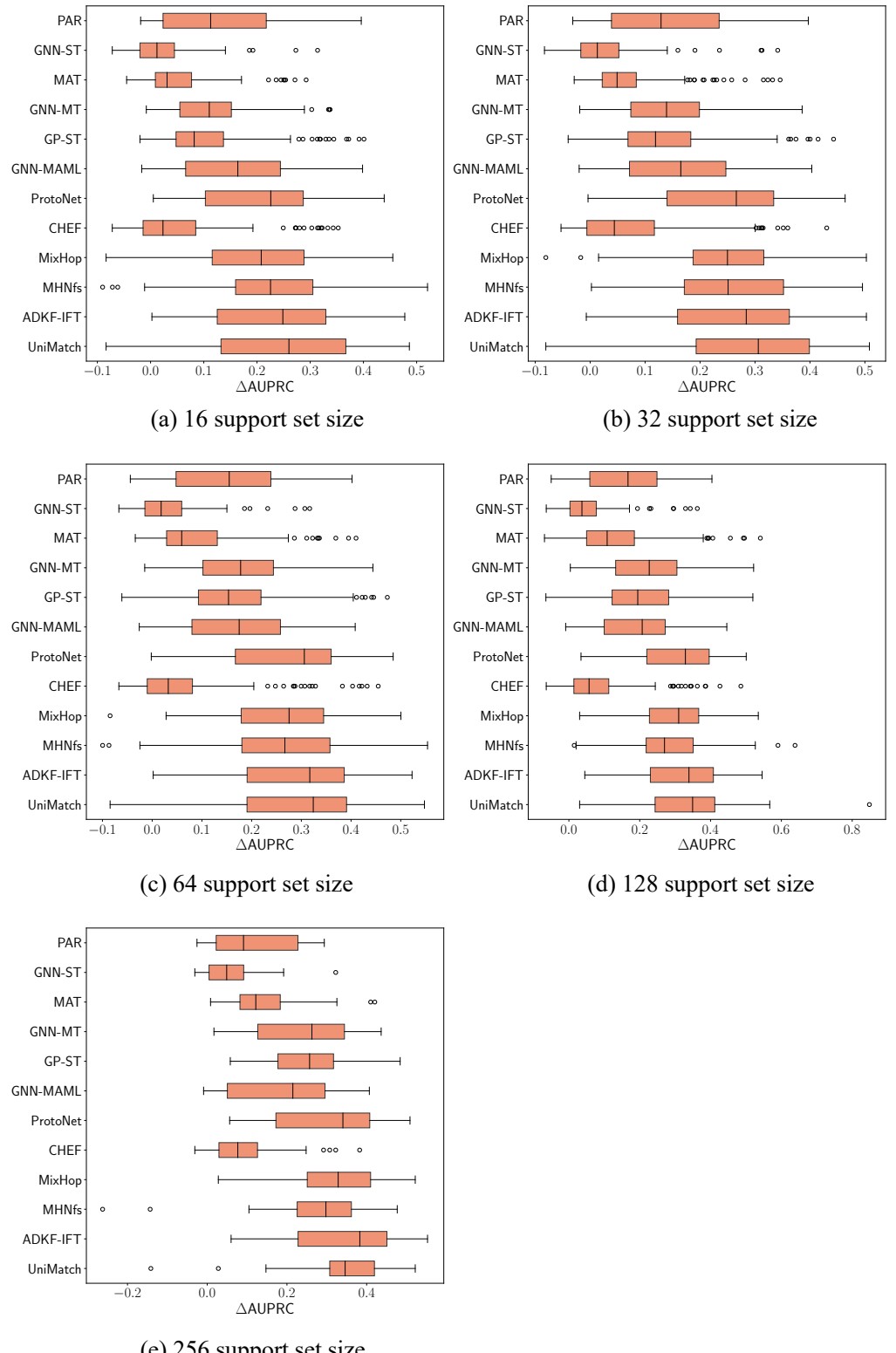

Figure 6: Box plots illustrate how different methods perform in classifying 157 FS-Mol test tasks across various support set sizes.

### E.3 META-TESTING COSTS

In this section, we compare the inference time of our UniMatch with meta-learning approaches. Figure 7 illustrates that UniMatch takes slightly more time compared to ProtoNet (Snell et al., 2017) and GNN-MAML (Guo et al., 2021). Additionally, ADKF-IFT (Chen et al., 2023) exhibits the longest reference time. However, it is important to note that UniMatch still maintains a relatively fast inference time, making it a viable option for meta-learning tasks.

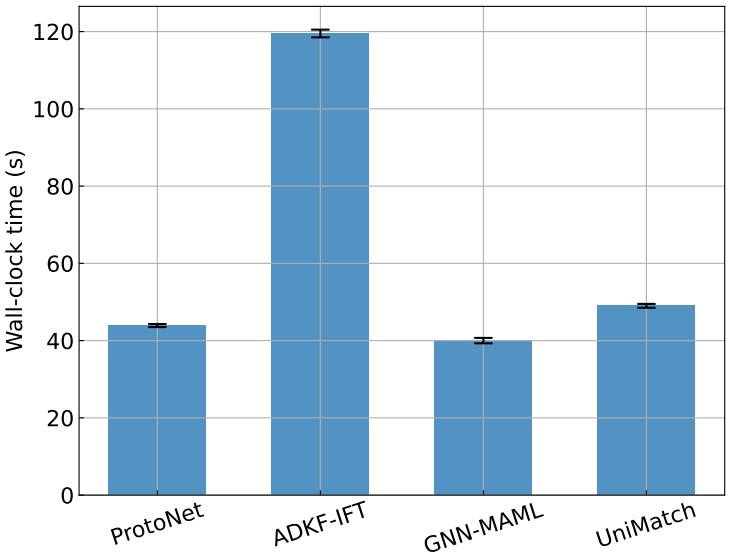

Figure 7: The wall-clock time, along with standard errors, is recorded during meta-testing on a predetermined set of FS-Mol classification tasks for comparison with the meta-learning approaches.

## F VISUALIZATION EXPERIMENTS

The details of the ten molecules used in Section 4.4 are represented in Table 6.

Table 6: The ten molecular sampled from the Tox21 dataset are used for the NR-AhR toxicity prediction task. "1" indicates active molecules, while "0" indicates inactive molecules.

| SMILES | Label |
| --- | --- |
| CCOc1ccc2nc(S(N)(=O)=O)sc2c1 | 1 |
| CC1=C(C(=O)Nc2ccccc2)S(=O)(=O)CCO1 | 0 |
| CC(C)(C)C1CCC(=O)CC1 | 0 |
| Nc1ccc(/N=N/c2ccccc2)cc1 | 1 |
| COCC(C)O | 0 |
| Nc1ccccc1C(=O)Oc1ccc2ccccc2c1 | 1 |
| ONc1ccccc1 | 1 |
| CC(O)CNCC(C)O | 0 |
| CCCCC(CC)CCC(CC(C)C)OS(=O)(=O)[O-] | 0 |
| O=C([O-])COc1nn(Cc2ccccc2)c2ccccc12 | 0 |

## G RELATED WORK

### G.1 HIERARCHICAL REPRESENTATION LEARNING ON GRAPHS

Hierarchical representation learning is crucial for graphs as it captures multi-scale structures, enabling models to discern both local and global patterns more effectively (Grattarola et al., 2022).

Existing methods can be divided into three categories: layer-wise hierarchical methods, architecture-level methods, and supergraph-based methods. Layer-wise hierarchical methods, such as DiffPool (Ying et al., 2018), JK-Nets (Xu et al., 2018), Top-K Pooling (Lee et al., 2019), ASAP (Ranjan et al., 2020), HGP-SL (Zhang et al., 2019), and MixHop (Abu-El-Haija et al., 2019), primarily capture graph structures by clustering nodes or using adjacency matrix powers, but they struggle to balance local and global structures, limiting their ability to capture multi-level features. Additionally, architecture-level methods, such as FraGAT (Zhang et al., 2021b), MGSSL (Zhang et al., 2021a), and UniCorn (Feng et al., 2024) capture multi-scale features but struggle with task-specific structural changes and often lack generalization, particularly for new tasks. Supergraph methods, like HiMol (Zang et al., 2023), offer hierarchical representation, but they are hindered by high computational complexity and insufficient sensitivity to structural diversity. To solve this issue, HiPM (Kang et al., 2024) introduces a prompt tree to model relationships between tasks. However, this approach still requires a substantial amount of data to be effective.

## H  DISCUSSION, LIMITATION AND FUTURE WORK

**Conclusion.**   In this paper, we propose Universal Matching Network (UniMatch) to address the limitations of existing few-shot learning approaches in drug discovery. UniMatch leverages a dual matching mechanism that integrates hierarchical molecular matching and implicit task-level matching to capture multi-scale structural features and inter-task relationships effectively. By utilizing hierarchical pooling and matching techniques, UniMatch aligns representations across atomic, sub-structural, and molecular levels, preserving essential structural details that are often overlooked by single-scale methods. Our experimental results demonstrated that UniMatch outperforms state-of-the-art methods on the MoleculeNet and FS-Mol benchmarks, with significant improvements in AUROC and $\Delta$AUPRC. Additionally, UniMatch showed exceptional generalization ability on the Meta-MolNet benchmark. However, our analysis revealed that the model's performance on regression tasks could be further improved by addressing specific issues in the fusion module.

**Limitation: Simple Fusion Design.**   The fusion mechanism in the proposed UniMatch model is relatively simplistic, which may limit its ability to effectively integrate information from different hierarchical levels. This design could result in suboptimal performance, as the model may not fully capture complex interactions and dependencies across multiple scales of molecular structures. More sophisticated fusion strategies, such as attention-based fusion or multi-scale feature aggregation, could enhance the model's capability to combine features more effectively. Leveraging such advanced techniques would enable UniMatch to exploit the rich hierarchical information inherent in molecular structures, thereby improving prediction accuracy and generalization. Although these advanced fusion methods may increase computational complexity, the potential gains in model performance justify this investment.

**Limitation: Underfitting on Regression Tasks.**   Our UniMatch model exhibits underfitting on regression tasks, indicating that it may not be capturing all the necessary features and complexities required for accurate predictions. Experimental results show that the underfitting issue arises from using a linear layer in the final fusion module. When different layer features are combined using a weighted average approach instead, the model achieves significantly better performance and converges properly. This suggests that the linear layer may not adequately capture the relationships between features from different layers for regression tasks. Therefore, replacing the linear fusion with a weighted average aggregation method could resolve the underfitting issue and allow the model to better capture complex feature relationships, thereby improving its performance on regression tasks.

**Future Work.**   In the future, we will focus on enhancing the fusion mechanism within UniMatch to better capture the complex relationships between features from different hierarchical levels. Specifically, we will explore advanced fusion techniques such as attention-based fusion and multi-scale feature aggregation to replace the current simplistic linear approach. Additionally, we plan to conduct more extensive experiments on a wider range of datasets and tasks to ensure the robustness and generalizability of our model. Another promising direction is to integrate domain-specific knowledge and features into the model to further improve its predictive accuracy. As part of our ongoing efforts, we also plan to further investigate the interpretability of UniMatch by incorporating

gradient-based methods (e.g., DeepLIFT (Shrikumar et al., 2017)) and exploring the Kolmogorov-Arnold Network (KAN) (Liu et al., 2024b) to gain deeper insights into feature importance and model decision-making. Finally, we will work on optimizing the computational efficiency and scalability of UniMatch to facilitate its application in large-scale drug discovery projects.

