# OpenReview forum: "UniMatch: Universal Matching from Atom to Task for Few-Shot Drug Discovery"
_ICLR.cc/2025/Conference — ICLR 2025 Spotlight_

### Official Review · Reviewer_E3W3 · 2024-10-18

**Soundness:** 2
**Presentation:** 3
**Contribution:** 3
**Rating:** 6
**Confidence:** 2

**Summary:**

This work presents UniMatch for molecular property prediction in the few-shot learning scenario. It combines explicit hierarchical molecular matching with implicit task-level matching via meta-learning. The effectiveness of the few-shot learning and multi-task generalization is validated on the MoleculeNet, FS-Mol, and Meta-MolNet benchmarks. An ablation study and visualizations are provided to demonstrate the importance of hierarchical molecular matching and task-level matching.

**Strengths:**

1. The writing is clear, with well-crafted figures that enhance understanding. It also offers a discussion of limitations and detailed supplementary explanations.

2. The experiments are quite extensive. The research examines the universality of the method across different network structures. It also conducts evaluations on three benchmarks. The results are convincing.

**Weaknesses:**

1. **Timeliness**:
 - The baselines and related work employed in the paper lack more recent works, especially those of recent 1 or 2 years.

- The pretraining model, Pre-GNN (Hu et al., 2020), is relatively old in the context of molecular pretraining. Recent years have seen the emergence of far more powerful models. I encourage authors to consider integrating more up-to-date pretraining models to showcase the significance and practical usage of their approach.

By simple searching, I find more recent related works on Hierarchical molecular representation learning methods like "UniCorn: A Unified Contrastive Learning Approach for Multi-view Molecular Representation Learning" (ICML 2024) and "Adapting Differential Molecular Representation with Hierarchical Prompts for Multi-label Property Prediction" (Briefings in Bioinformatics, 2024).



2. **Novelty**:

The hierarchical molecular representation extraction concept is commonly seen in the literature. Additionally, the paper employs standard attention and meta-learning methods. I'm concerned that this may not meet the novelty standards of top conferences.

**Questions:**

1. **Generalization to different molecular structures**:
It would be beneficial to understand how well the method generalizes to diverse molecular structures, as this is crucial for practical applications. For example, testing under a scaffold split for support and query sets could provide insights. The paper mentions the method's failure on MUV. Is this due to a lack of structure generalization? The phrase "severe distribution imbalance" is not clearly explained. What does it refer to specifically? And why are other baselines able to overcome this issue?

2. In Figure 2, the query molecule appears identical to one in the support set. In your implementation, can the query molecule overlap with the support set?

3. The numbers in parentheses following the datasets in Table 1, which appear to represent the number of tasks, do not seem to be explained in the main text.

---

### Official Review · Reviewer_6vhN · 2024-10-24

**Soundness:** 4
**Presentation:** 2
**Contribution:** 3
**Rating:** 8
**Confidence:** 4

**Summary:**

This study introduces a new framework for molecular representation specially aimed at multi-task few-shot learning. In this context for each task there is a small support subset of molecules with known labels ($S_T$) and a query subset ($Q_T$) with unknown labels.

The contributions of the study can be broken down into the following components:

1. Matching at different levels of the molecular graph. This is achieved by an attention mechanism where the keys are the molecular features of the $S_T$, the molecular representations of $Q_T$ are used as query, and as value, the labels of the $S_T$. This matching attention mechanism is not novel. The novelty resides on repeating this matching with the representations at each layer of the GNN (thus effectively capturing information at different structural levels of any given molecule) and concatenating the resulting values that are then passed through an MLP to predict the final target value.
2. Meta-learning. An additional implicit matching mechanism is used to learn the similarity between different tasks and how to leverage the multi-scale representations from one task to another. This is achieved by an inner loop where the task-specific adaption parameters are updated using the matrix that captures the similarity between tasks and an outer loop where this similarity matrix is updated.

The results obtained in benchmark datasets clearly show that the benefits of this approach when compared to current state-of-the-art alternatives. The ablation study also clearly supports the contribution of all the components of the method.

**Strengths:**

1. The paper tackles a valuable domain-specific problem, which high societal value, with a method that can be reasonably adapted to different domains.
2. The evaluation is extensive and clearly demonstrates the advantages of the method.
3. The ablation study is complete and clearly demonstrates the contribution of every component in the method.

**Weaknesses:**

Major weaknesses
---
1. The technical description of the dual matching mechanism is hard to follow and it is a disservice to an otherwise solid work.


Minor weaknesses
---
_(These are minor points that do not have a direct bearing in my evaluation of the paper, but I think would improve its quality)_

1. In line 075-077 and Figure 1, one of the examples provided of the importance of the multi-level representation of molecules from atomic, to whole molecule serve their purpose as an illustration, but it is not chemically correct. Hydrogen ions do not exist within the molecule, they would be referred to as hydrogen atoms. Though it is true that Hydrogens ions in a solution determine its acidity. In a molecule, the hydrogen atoms are not the determinants of acidity, but rather the atoms they are attached to. So in the first examples, the acidity of those molecules is not determined by the Hydrogen atoms but by the chlorine and oxygen atoms; and their electronegativity (or ability to hold onto the Hydrogen). Still the illustration is accurate, but the atom that should be highlighted is not the Hydrogen. In the case of the right example, it is arguable that the acidity is not only caused by the oxygen, but rather the whole substructure of H-O-S.
2. In line 409, when describing the FS-Mol dataset, it is described as "[a benchmark] for macromolecules (i.e., proteins)". This is not accurate, the benchmark measures the ability of small molecules (the one that serve as input for this system) to bind to proteins. In other words, the tasks are whether the molecules can bind to specific proteins. This is an important distinction because due to their size, modelling proteins with the approach presented in this paper would be computationally infeasible.

**Questions:**

1. When evaluating against MoleculeNet tasks, is there a reason why the QM tasks were not considered and only the biophysical properties are included?
2. Do you have any ideas as to whether more explicitly introducing an inductive bias to the different levels of attention could be helpful?

---

### Official Review · Reviewer_PgJj · 2024-11-03

**Soundness:** 3
**Presentation:** 4
**Contribution:** 3
**Rating:** 8
**Confidence:** 4

**Summary:**

The paper presents UniMatch, a framework designed to address data scarcity in few-shot drug discovery. It uses a dual matching approach: explicit hierarchical molecular matching, which captures information from the atomic level to full molecular structures, and implicit task-level matching via meta-learning to enable effective generalization across tasks.

**Strengths:**

1. The concept of modeling molecular structures across multiple levels is promising.
2. The manuscript is well-written and easy to read.
3. Extensive experiments across diverse datasets demonstrate that the model consistently outperforms baselines.

**Weaknesses:**

1. The related work section is somewhat lengthy. It might be more effective to move it to the end of the manuscript or condense it.
2. While the rationale for modeling hierarchical structures is solid, I have concerns about the use of mean pooling. This approach may lead to a predominantly molecule-level representation, potentially losing crucial atomic and substructural details. What do the authors think?

**Questions:**

see weaknesses

---

### Official Review · Reviewer_XF6i · 2024-11-04

**Soundness:** 3
**Presentation:** 4
**Contribution:** 3
**Rating:** 8
**Confidence:** 3

**Summary:**

The authors propose UniMatch, a dual matching framework that integrates hierarchical molecular matching with meta-learning, significantly improves the performance and generalization in molecular property prediction tasks.

**Strengths:**

1.  UniMatch is the first method that integrates multiple levels of molecular structures, demonstrating outstanding performance across various tasks and sigificantly contributes to the filed of drug discovery.
2. This paper is clearly-written and accompanied supported by well-designed figures, well done!

**Weaknesses:**

1. I suggest further investigating the interpretability of the model, particularly with respect to the multi-level representations. Some gradient-based methods (e.g. DeepLIFT) could be used to reveal the importance of these features.

**Questions:**

1. Do AUCPR and AUPRC refer to the same metric? If so, I sugget using a consistent abbreviation throughout the paper.

---

### Author Response · Authors · 2024-11-28
**A Summary of Our Contributions and Responses**

Dear Reviewers and ACs,

As the discussion is ending in the next few hours, we are grateful for the insightful comments and valuable suggestions from all reviewers.  In the following, we would like to summarize the contributions and responses of this paper again.


**Contributions**:
- To tackle molecular property prediction in few-shot scenarios, we propose a unified matching framework that combines hierarchical molecular matching with task-level matching through meta-learning. This approach is **effective** (R XF6i), **innovative** (R XF6i), **promising** (R PgJj), and **adaptable to various domains with high social value** (R 6vhN).
- The experiments are **extensive** (R PgJj, R 6vhN, R E3W3), covering three benchmarks, and the ablation study is **complete** (R 6vhN).
- The paper is **well-written** (R  XF6i, R PgJj, R E3W3), and the figures are **well-crafted** (R XF6i, R E3W3).

**Responses and Revisions**:
- For Reviewer XF6i's concerns:

   - **Q1 Further improving model interpretability**: We analyzed the matching scores and parameters of each layer, showing their complementary roles in decision-making. We also plan to further enhance interpretability using DeepLIFT and the Kolmogorov-Arnold Network (KAN).

   - **Q2 AUCPR and AUPRC**: To avoid confusion, we have standardized the terminology to "AUPRC" throughout the paper.

- For Reviewer PgJj's concerns:

  - **Q1 Mean pooling may result in the loss of details**: We compared mean pooling with attention-based and ensemble-based pooling. The latter methods perform better at preserving fine-grained details. We will incorporate these advanced pooling strategies in future work.

  - **Q2 Length of the related work section**: We agree with the suggestion to condense the related work section and move it to the end of the manuscript. This change will improve the paper’s flow and readability.

- For Reviewer 6vhN's concerns:

   - **Q1 More QM tasks in MoleculeNet**: We’ve added QM9 experiments to our evaluation, where UniMatch achieves the best performance with the lowest MAE.

   - **Q2 More idea about Inductive bias in attention**: We plan to incorporate structural priors at different levels (atomic, substructural, molecular) to guide attention and improve performance.

  - **Q3 Clarifying dual matching mechanism**: The dual matching mechanism combines hierarchical molecular matching (using GNN and attention for multi-level comparisons) and task-level matching (using meta-learning for task adaptation). A detailed explanation and algorithm are now in the appendix A.

  - **Q4 Minor issues**: We’ve corrected Figure 1 (HF and NH₃) and updated the FS-Mol description in the paper.

- For Reviewer E3W3's concerns:

   - **Q1 Outdated baselines**: We have clarified the choice of standard baselines and updated the related work to include recent studies like UniCorn and HiPM.

  - **Q2 More Pretraining models**: We have incorporated the advanced GraphMAE pretraining model, achieving significant performance improvements.

   - **Q3 Novelty concerns**: We have emphasized our task-matching mechanism as a unique contribution to few-shot learning in molecular representation.

   - **Q4 Generalization and dataset imbalance**: We have analyzed the extreme class imbalance in the MUV dataset and demonstrated generalization through scaffold-based splits.

   - **Q5 Minor issues**: We have corrected the query molecule in Figure 2 and clarified the meaning of numerical values in Table 1.


Warm regards,
The Authors of Submission 45

---

### Public Comment · ~Kun_Li10 · 2026-01-17
**Where is Code?**

Your code is None in Github

---

> ### Public Comment · ~Ruifeng_Li1 · 2026-01-17
>
> Thank you very much for your attention to our work. We plan to release our code in the next one to two weeks.

---

> > ### Public Comment · ~Yuanpeng_Zhang3 · 2026-04-01
> >
> > I'm very sorry to bother you, but your GitHub repository is still empty at this time.

---

### Meta-Review · Area_Chair_Khsh · 2024-12-16

**Metareview:**

In this submission, the authors propose an effective method for few-shot drug discovery. The proposed method leverages the multi-scale representation of molecules and connects it with implicit task-level generalization, which provides the model performance in various datasets. The proposed method fully leverages the hierarchical structure of molecules and thus makes the representations at different levels work for various tasks, which is reasonable and effective. All the reviewers appreciate this work and are satisfied with the authors' feedback in the rebuttal phase. Taking all these into account, I tend to accept this work.

**Additional Comments On Reviewer Discussion:**

All the reviewers scored this submission positively. Three of the four reviewers interacted frequently with the authors. The authors' responses were effective and resolved the reviewers' concerns successfully.

---

### Decision · Program_Chairs · 2025-01-22

Accept (Spotlight)